# A Performance–Consumption Balanced Scheme of Multi-Hop Quantum Networks for Teleportation

**Jin Xu** [ID]**, Xiaoguang Chen \*, Hanwei Xiao, Pingxun Wang and Mingzi Ma** [ID]

Department of Communication Science and Engineering, Fudan University, Shanghai 200433, China; 19210720064@fudan.edu.cn (J.X.); 19210720073@fudan.edu.cn (H.X.); wangpingxun0209@163.com (P.W.); 19210720078@fudan.edu.cn (M.M.)

\* Correspondence: xgchen@fudan.ac.cn or xiaoguangchen@fudan.edu.cn

**Abstract:** Teleportation is an important protocol in quantum communication. Realizing teleportation between arbitrary nodes in multi-hop quantum networks is of great value. Most of the existing multi-hop quantum networks are based on Bell states or Greeberger–Horne–Zeilinger (GHZ) states. Bell state is more susceptible to noise than GHZ states after purification, but generating a GHZ state consumes more basic states. In this paper, a new quantum multi-hop network scheme is proposed to improve the interference immunity of the network and avoid large consumption at the same time. Teleportation is realized in a network based on entanglement swapping, fusion, and purification. To ensure the robustness of the system, we also design the purification algorithm. The simulation results show the successful establishment of entanglement with high fidelity. Cirq is used to verify the network on the Noisy Intermediate-Scale Quantum (NISQ) platform. The robustness of the fusion scheme is better than the Bell states scheme, especially with the increasing number of nodes. This paper provides a solution to balance the performance and consumption in a multi-hop quantum network.

**Keywords:** multi-hop quantum networks; teleportation; fusion; entanglement purification

## 1. Introduction

Teleportation is always used for transferring quantum information with the aid of maximally entangled states and classical channels. In 1993, Bennet proposed the first teleportation protocol based on the Einstein–Podolsky–Rosen (EPR) state [1]. In 1997, Bouwmeester proved it in the practical experiment [2]. Since then, many other teleportation protocols were proposed based on different entangled states, such as GHZ states, W states, and cluster states [3–5].

Teleportation can be easily realized between two adjacent nodes. However, in a multi-hop quantum network, most nodes are not directly connected. Realizing teleportation in a multi-hop quantum network is valuable. In 2005, Cheng proposed the Bridging protocol of the network in this field [6], and then, the ideas for teleportation network were given in different entangled states [7,8]. However, they are all designed based on the teleportation between directly connected nodes. Chen gave a classical solution to it [9]. Quantum states in all intermediate nodes are measured in the end to realize teleportation. Increasing the number of nodes between the source and the destination will bring difficulties with exponential growth to the calculation. Besides, the influence of noise is also amplified.

As an important technology in quantum relay, entanglement swapping has been realized in the recent years [10–13]. The original entanglement swapping protocol took advantage of the spontaneous parametric down-conversion (SPDC) source [14]. Today, scientists prefer to use hybrid protocols and photon interference to entangle spins or atoms [15,16]. Scientists have proved that entanglement swapping can establish entanglement between states in different entangled pairs. The price is the loss of fidelity [17,18]. This technology makes it possible for arbitrary nodes in the network to share entangled

states. When it comes to the noise in the environment, scientists think that purification can solve this problem [19–21]. Because of a developed purification [22–25], high fidelity entanglement can be established in the multi-hop quantum network.

Based on the entanglement swapping and purification, we design a new scheme of a multi-hop quantum network. Taking advantage of entanglement swapping and purification, nodes in the network can share entangled states with high fidelity. To ensure the high robustness of the system, we design a purification algorithm. To ensure the practicability of the system, we use bit-flip, phase-flip, and depolarizing noise models in the simulation. Although amplitude damping noise seems more practical in the simulation, we find that it will lead to the disappearance of entanglement. Considering purification always works on entangled states, thus, we did not use amplitude damping noise in this paper. Besides, we consider the decline in fidelity of entanglement swapping based on the latest physical experiment. The number of entering nodes is decreased in the final communication to protect the privacy. Thus, the computational complexity is reduced. In the end, we simulate the network on Cirq with bit-flip and depolarizing noise in the true environment.

Contributions of this paper can be stated as follows:

- This paper proposes a new scheme of a multi-hop quantum network based on bipartite communication with fusion states. It improves the noise immunity of the network while avoiding the huge consumption.
- A purification algorithm is proposed to ensure the robustness of the system, providing a solution for future quantum communication.
- The fidelity for multi-particle entanglement purification is calculated.

The paper is outlined as follows. In Section 2, we review entangled states, teleportation, entanglement swapping, quantum noise, and fidelity. In Section 3, we design the new multi-hop quantum network architecture based on bipartite communication. Besides, we give the purification scheme for the Bell pair and GHZ states. In Section 4, we give the design for a purification algorithm and simulation results. Finally, conclusions are given in Section 5.

## 2. Preliminary

### 2.1. Entangled State

The Bell state is the most commonly used entangled state in quantum communication. The basic state in this paper is

$$|0\rangle = \begin{pmatrix} 1 \\ 0 \end{pmatrix}, |1\rangle = \begin{pmatrix} 0 \\ 1 \end{pmatrix}. \tag{1}$$

Then, the Bell states can be expressed as

$$\begin{aligned}
\left|\Phi^+\right\rangle &= \frac{1}{\sqrt{2}}(|00\rangle + |11\rangle), \left|\Phi^-\right\rangle = \frac{1}{\sqrt{2}}(|00\rangle - |11\rangle) \\
\left|\Psi^+\right\rangle &= \frac{1}{\sqrt{2}}(|01\rangle + |10\rangle), \left|\Psi^-\right\rangle = \frac{1}{\sqrt{2}}(|01\rangle - |10\rangle).
\end{aligned} \tag{2}$$

If applying a unitary operation such as the Pauli matrix on Bell states, we obtain

$$\begin{aligned}
\left|\Psi^+\right\rangle &= (X \cdot |0\rangle) \otimes |0\rangle + (X \cdot |1\rangle) \otimes |1\rangle = X \cdot \left|\Phi^+\right\rangle \\
\left|\Phi^-\right\rangle &= (Z \cdot |0\rangle) \otimes |0\rangle + (Z \cdot |1\rangle) \otimes |1\rangle = Z \cdot \left|\Phi^+\right\rangle \\
\left|\Psi^-\right\rangle &= (Y \cdot |0\rangle) \otimes |0\rangle + (Y \cdot |1\rangle) \otimes |1\rangle = Y \cdot \left|\Phi^+\right\rangle,
\end{aligned} \tag{3}$$

where

$$X = \begin{pmatrix} 0 & 1 \\ 1 & 0 \end{pmatrix}, Y = \begin{pmatrix} 0 & -i \\ i & 0 \end{pmatrix}, Z = \begin{pmatrix} 1 & 0 \\ 0 & -1 \end{pmatrix}.$$

Thus, we choose $|\Phi^+\rangle$ to share in the network. Another entangled state commonly used in quantum communication is the GHZ state. Since the fidelity of GHZ states decreases with the increasing number of qubits [25], we select 3-qubit GHZ states to use.

$$
\begin{aligned}
|\Psi_1\rangle &= \frac{1}{\sqrt{2}}(|000\rangle + |111\rangle), |\Psi_2\rangle = \frac{1}{\sqrt{2}}(|001\rangle + |110\rangle) \\
|\Psi_3\rangle &= \frac{1}{\sqrt{2}}(|010\rangle + |101\rangle), |\Psi_4\rangle = \frac{1}{\sqrt{2}}(|011\rangle + |100\rangle) \\
|\Psi_5\rangle &= \frac{1}{\sqrt{2}}(|000\rangle - |111\rangle), |\Psi_6\rangle = \frac{1}{\sqrt{2}}(|001\rangle - |110\rangle) \\
|\Psi_7\rangle &= \frac{1}{\sqrt{2}}(|010\rangle - |101\rangle), |\Psi_8\rangle = \frac{1}{\sqrt{2}}(|011\rangle - |100\rangle).
\end{aligned}
\tag{4}
$$

For the same reason, we choose $|\Psi_1\rangle$ to share in the quantum network.

### 2.2. Teleportation

Teleportation is a safe communication method when facing eavesdropper. It sends only measurement results in a classical channel, instead of sending the quantum state directly. The receiver can rebuild the quantum state based on the measurement results. Even if the classical channel is eavesdropped, qubit still cannot be rebuilt because of the lacking entangled state.

Teleportation based on the Bell pair is shown in Figure 1. Nodes A and B share a Bell pair $|\Phi^+\rangle$. Node A wants to send an arbitrary state $|\psi\rangle = \alpha|0\rangle + \beta|1\rangle$ to node B, where $|\alpha|^2 + |\beta|^2 = 1$. The control state and the entangled state in node A are measured by the circuit shown in Figure 1. State $|\psi\rangle$ can be reconstructed in node B by a quantum gate based on the entangled state. Quantum gate matching to the measurement results is shown in Appendix A.

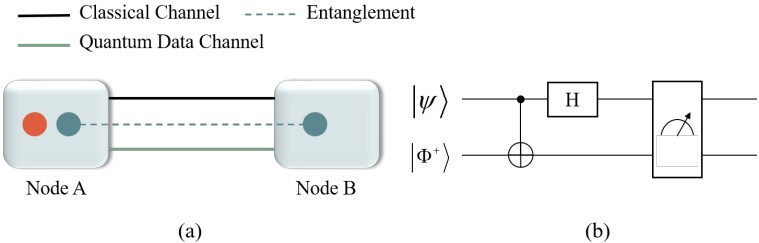

(a)                                    (b)

**Figure 1.** (**a**) Entangled states in teleportation model and (**b**) circuit in node A based on Bell pair. Classical channel is used to transfer measurement results, quantum channel is for entanglement distribution. Here, H represents the Hadamard gate. $|\psi\rangle$ is the control qubit of the CNOT gate.

Teleportation based on GHZ states is shown in Figure 2. Node A holds two entangled states and an arbitrary state $|\psi\rangle$.

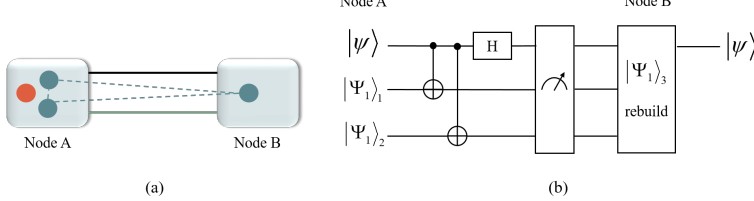

(a)                                    (b)

**Figure 2.** (**a**) Teleportation model and (**b**) circuit based on GHZ states. GHZ states $|\Psi_1\rangle_i$, $i = 1, 2, 3$ are shared between nodes A and B. Node A holds three qubits operated by CNOT and the H gate in the circuit, and sends measurement results to node B to rebuild $|\Psi\rangle$ based on $|\Psi_1\rangle_3$.

### 2.3. Entanglement Swapping

In a multi-hop quantum network, entanglement swapping can establish the entanglement between nodes without being directly connected. So, it is also an important technology to realize a quantum relay [13].

Node B has shared entanglement both with nodes A and C, as shown in Figure 3. Entanglement swapping can help to establish entanglement between node A and C. It is described in detail in Appendix B.

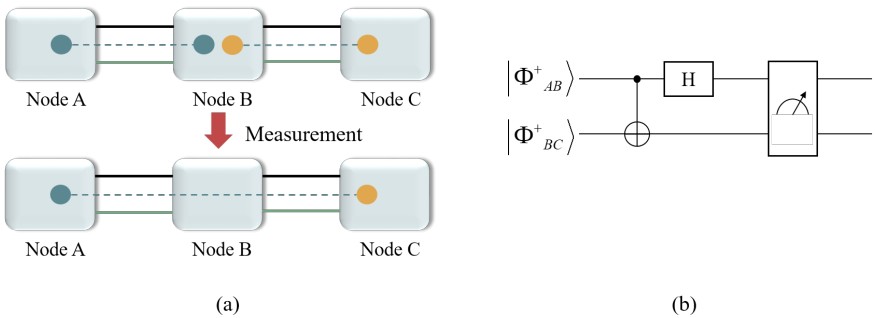

(a)　　　　　　　　　　　　　　　　　(b)

**Figure 3.** (**a**) Entanglement swapping based on Bell pairs and (**b**) circuit in node B. $\left|\Phi_{AB}^{+}\right\rangle$ is shared between nodes A and B. $\left|\Phi_{BC}^{+}\right\rangle$ is shared between nodes B and C. Once two states in node B are measured, entanglement between nodes A and C is established.

### 2.4. Fidelity

Because of the channel noise, the quantum state we finally get may be different from the theoretical value. Fidelity is used to measure this deviation, defined in [26]:

$$F = \sqrt{\langle\psi|\rho|\psi\rangle},\tag{5}$$

where $|\psi\rangle$ is the wanted quantum state, $\rho$ is the density matrix of the quantum state received finally. The density matrix is

$$\rho = \sum_i p_i|\psi_i\rangle\langle\psi_i|,\tag{6}$$

where $p_i$ is the probability of state $|\psi_i\rangle$. Fidelity takes values from 0 to 1.

### 2.5. Noisy Channel

The main noisy channels in this paper are the bit-flip channel, phase-flip channel and depolarizing channel in [26]. The bit-flip channel flips a qubit from $|1\rangle$ to $|0\rangle$ with probability $p, 0 < p < 1$. Its operation elements are:

$$E_0 = \sqrt{1-p}I, E_1 = \sqrt{p}X.\tag{7}$$

For the phase-flip channel, the operation elements are:

$$E_0 = \sqrt{1-p}I, E_1 = \sqrt{p}Z.\tag{8}$$

For the depolarizing channel, with error probability $p$, the state can be described as

$$\varepsilon(\rho) = (1-p)\rho + \frac{p}{3}(X\rho X + Y\rho Y + Z\rho Z),\tag{9}$$

where $X$, $Y$, and $Z$ are Pauli operations.

There are also other noise models, such as Gaussian noise, but they are hard to realize on the NISQ platform. So, we choose the three most commonly used quantum noise models for the channel environment simulation. Since purification results under bit-flip and phase-

flip noise are similar, we only provide simulations under bit-flip and depolarizing noise in Section 4.

## 3. The Scheme of Network Architecture and Purification

### 3.1. Network Architecture Based on Bipartite Communication

For a network based on Bell states, a Bell pair is pre-shared between adjacent nodes. Assume that A is the source node while E is the destination node, and they are not directly connected. Entanglement swapping can establish entanglement between them. However, physical experiment has proved that each swapping will bring about 12.77 ± 4.17% loss of entanglement fidelity [17,18]. The number of times a qubit taking part in entanglement swapping should be as small as possible. Node A can select the node between the source and destination for entanglement swapping through transferring commands in the classical channel.

Compared with the Bell state, a multi-particle state such as the GHZ state has a better entanglement property [27,28]. Therefore, the GHZ state is more widely used in teleportation. However, once we measure one particle of GHZ states, the whole entanglement will be destroyed. To solve this problem in bipartite communication, fusion is used to form GHZ states based on Bell pairs [25]. The principle and circuit of fusion are shown in Figure 4.

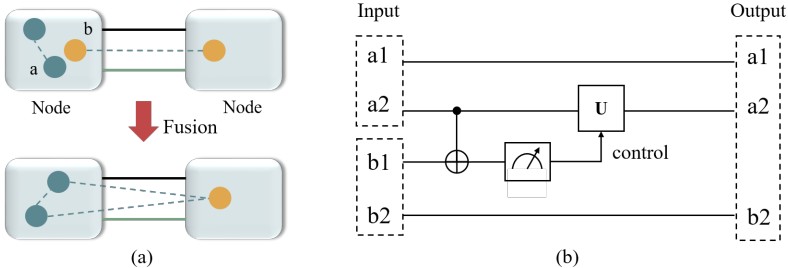

**Figure 4.** (**a**) Fusion principle and (**b**) circuit. Assume that nodes A and B share a pair of Bell states b1 and b2, and node A holds the other Bell pair a1 and a2. Circuit in (**b**) can be applied in two entangled particles a2 and b1. One only needs measure b1 in the Z basis and apply the Pauli gate correction on a2 based on the measurement result. Then, a1, a2 and b2 are changed into GHZ states.

The fusion scheme in the 5-node network is shown in Figure 5. The first round of entanglement swapping in the network occurs in nodes B and D, while the second round occurs in node C. Thus, entanglement is established between nodes A and E.

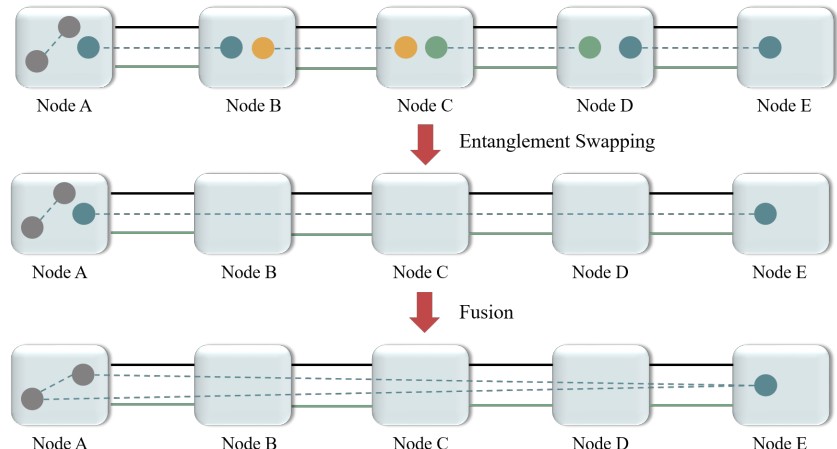

**Figure 5.** Entanglement swapping and fusion in a quantum network. Entanglement swapping can help to establish entanglement between nodes A and E based on Bell states. Fusion in node A can help to form GHZ states.

### 3.2. Purification Circuit for Different States

The accurate realization of teleportation needs high-fidelity entangled states. Purification is a common method to improve the fidelity.

#### 3.2.1. Purification of Bell Pairs

The standard entanglement purification protocol (EPP) is the way to reduce the influence of bit-flip noise. A Bell pair is measured to improve the fidelity of another pair. To solve the phase-flip noise, quantum privacy amplification (QPA) was proposed in [29], shown in Figure 6.

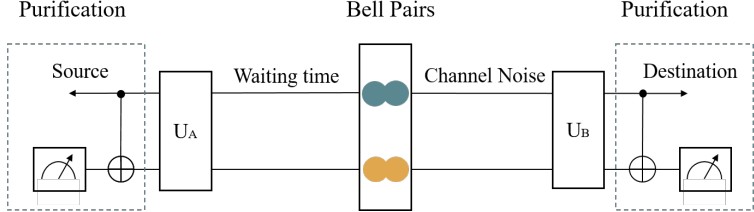

**Figure 6.** Purification circuit for a Bell pair. Assume two Bell pairs are distributed to the source and the destination node. $U_A$ and $U_B$ are rotation operators to solve phase-flip noise. Fidelity of one entangled pair can be improved by measuring the other entangled pair.

$U_A$ and $U_B$ are rotation operators for $\theta$ about x-axes, noted as $R_x(\theta)$ [30].

$$R_x(\theta) = e^{-j\theta X/2} = \cos\frac{\theta}{2}I - j\sin\frac{\theta}{2}X, \tag{10}$$

where $\theta$ is $-\pi/2$ for $U_A$, and $\pi/2$ for $U_B$. Assume that the two Bell pairs are both $|\Phi^+\rangle$ with the fidelity of 1 at first. After quantum distribution, $U_A$ acts on the quantum state in the source node, while $U_B$ acts on the state in the destination node.

For bit-flip noise, assume that the error probability of each qubit is $p$, where $0 < p < 0.5$. Let $q = 1 - (1-p)^2 - p^2$. The two nodes share $|\Phi^+\rangle$ with the probability $1 - q$. The Bell pairs pre-shared by two nodes with probability are shown in Table 1.

**Table 1.** Shared Bell pairs and the matching probability. Here, $q = 1 - (1-p)^2 - p^2$.

| One Shared Pair | Probability | Two Shared Pairs | Probability |
|---|---|---|---|
| $\|00\rangle + \|11\rangle$ | $(1-p)^2$ | $\|\Phi_c^+\rangle \otimes \|\Phi^+\rangle$ | $(1-q)^2$ |
| $\|10\rangle + \|01\rangle$ | $p(1-p)$ | $\|\Phi_c^+\rangle \otimes \|\Psi^+\rangle$ | $(1-q)q$ |
| $\|01\rangle + \|10\rangle$ | $(1-p)p$ | $\|\Psi_c^+\rangle \otimes \|\Phi^+\rangle$ | $q(1-q)$ |
| $\|11\rangle + \|00\rangle$ | $p^2$ | $\|\Psi_c^+\rangle \otimes \|\Psi^+\rangle$ | $q^2$ |

$|\Phi^+\rangle_c$ is the control pair, the other is target pair. Prepare two Bell pairs $|\Phi^+\rangle_c \otimes |\Phi^+\rangle$ at first. Because of the noise, four possible scenarios will thus occur for the two Bell pairs before purification. We can obtain the final results as

$$
\begin{aligned}
|\Phi^+\rangle_c \otimes |\Phi^+\rangle &\xrightarrow{CNOT} |\Phi^+\rangle_c \otimes |\Phi^+\rangle \xrightarrow{Measure} |\Phi^+\rangle_c \\
|\Phi^+\rangle_c \otimes |\Psi^+\rangle &\xrightarrow{CNOT} |\Phi^+\rangle_c \otimes |\Psi^+\rangle \xrightarrow{Measure} discard \\
|\Psi^+\rangle_c \otimes |\Phi^+\rangle &\xrightarrow{CNOT} |\Psi^+\rangle_c \otimes |\Psi^+\rangle \xrightarrow{Measure} discard \\
|\Psi^+\rangle_c \otimes |\Psi^+\rangle &\xrightarrow{CNOT} |\Psi^+\rangle_c \otimes |\Phi^+\rangle \xrightarrow{Measure} |\Psi^+\rangle_c.
\end{aligned}
\tag{11}
$$

If the measurement results of the target pair are equal, we save the control pair and judge purification to be successful. Otherwise, we discard the control pair with using the

measurement and prepare new pairs until the purification succeeds. Finally, we obtain $|\Phi^+\rangle_c$ with probability $\frac{(1-q)^2}{(1-q)^2+q^2}$.

Fidelity of entanglement states before purification is described as $F_{initial}$,

$$F^2_{initial} = (1-q)(\langle\Phi^+|\Phi^+\rangle)^2 + q\langle\Phi^+|\Psi^+\rangle\langle\Psi^+|\Phi^+\rangle = 1 - q. \tag{12}$$

Fidelity after purification is $F_{purify}$,

$$\begin{aligned}F^2_{purify} &= \frac{(1-q)^2}{(1-q)^2+q^2}\langle\Phi^+|\Phi^+\rangle\langle\Phi^+|\Phi^+\rangle + \frac{q^2}{(1-q)^2+q^2}\langle\Phi^+|\Psi^+\rangle\langle\Psi^+|\Phi^+\rangle \\ &= \frac{(1-q)^2}{(1-q)^2+q^2}.\end{aligned} \tag{13}$$

For phase-flip noise, the calculation and result are consistent thanks to the rotation operators.

### 3.2.2. Purification of GHZ States

A purification circuit for GHZ states is designed in Figure 7.

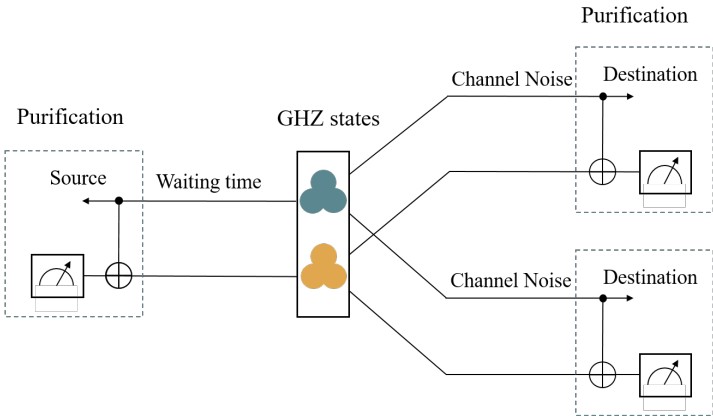

**Figure 7.** Purification circuit for GHZ states. Assume that two groups of 3-qubit GHZ states are distributed to the source and the destination node. No rotation operator exists here that can help to solve phase-flip noise, but the design is still useful for bit-flip noise. The circuit helps improve the fidelity of the control group by measuring the target group.

For bit-flip noise, assume bit-flip noise occurs with probability $p$, then $q = 1 - (1 - p)^3 - p^3$. The states shared before purification and the matching probability are shown in Table 2. $|\Psi_i\rangle_c$ is the control group in CNOT gate, the other is the target group.

**Table 2.** Shared GHZ states and the matching probability (bit-flip noise).

| Shared States | Probability |
|:---:|:---:|
| $|\Psi_1\rangle$ | $(1-p)^3 + p^3$ |
| $|\Psi_2\rangle$ | $p(1-p)^2 + p^2(1-p)$ |
| $|\Psi_3\rangle$ | $p(1-p)^2 + p^2(1-p)$ |
| $|\Psi_4\rangle$ | $p(1-p)^2 + p^2(1-p)$ |
| $|\Psi_1\rangle_c \otimes |\Psi_1\rangle$ | $(1-q)^2$ |
| $|\Psi_1\rangle_c \otimes |\Psi_i\rangle$ | $(1-q)q/3, i = 2,3,4$ |
| $|\Psi_i\rangle_c \otimes |\Psi_1\rangle$ | $q(1-q)/3, i = 2,3,4$ |
| $|\Psi_i\rangle_c \otimes |\Psi_j\rangle$ | $q^2/9, i,j = 2,3,4$ |

If the measurement results of the target group are equal, we save the control group and judge purification to be successful. Otherwise, we discard the control group and

prepare new groups until purification succeeds. After purification, we obtain $|\Psi_1\rangle_c$ with probability $\frac{(1-q)^2}{(1-q)^2+q^2/3}$.

The fidelity after purification is $F_{purify}$

$$
\begin{aligned}
F^2_{purify} &= \frac{(1-q)^2\langle\Psi_1|\Psi_1\rangle\langle\Psi_1|\Psi_1\rangle}{(1-q)^2+q^2/3} + \frac{q^2\langle\Psi_1|\Psi_2\rangle\langle\Psi_2|\Psi_1\rangle}{3(1-q)^2+q^2} \\
&+ \frac{q^2\langle\Psi_1|\Psi_3\rangle\langle\Psi_3|\Psi_1\rangle}{3(1-q)^2+q^2} + \frac{q^2\langle\Psi_1|\Psi_4\rangle\langle\Psi_4|\Psi_1\rangle}{3(1-q)^2+q^2} \\
&= \frac{(1-q)^2}{(1-q)^2+q^2/3}.
\end{aligned}
\tag{14}
$$

For phase-flip noise, a similar rotation operator does not exist. However, we can take advantage of quantum channel recognition by a quantum neural network [31]. If the noise is phase-flip, we can use the H gate to transform the noise into bit-flip before the transmission.

For n-particle GHZ states, if $p$ is the bit-flip probability, then $q = 1 - (1-p)^n - p^n$. The fidelity before purification is

$$
F^2_{initial} = 1 - q.
\tag{15}
$$

The fidelity after purification is

$$
F^2_{purify} = \frac{(1-q)^2}{(1-q)^2 + q^2/(2^{n-1}-1)}.
\tag{16}
$$

## 4. Design of Purification Algorithm and Simulation Results

### 4.1. Purification Algorithm

Purification is realized in a quantum network under the pumping manner as reported in [22] and shown in Figure 8.

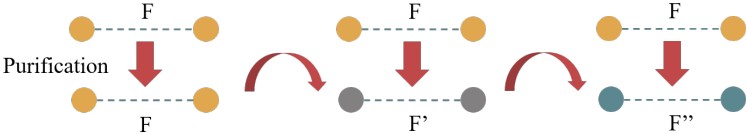

**Figure 8.** Entanglement pumping for purification. $F$ is the fidelity of the entangled states before purification, satisfying $F'' > F' > F$. Purification increases the fidelity of control states to $F''$. Target states are always with fidelity $F$.

The fidelity as a function of the error probability is shown in Figure 9. Purification time (PT) means the number of successful purifications. When PT = 0, Bell states are more resilient against noise than GHZ states. With the help of purification, we can see a huge improvement in fidelity. Under the existence of purification, lines of GHZ states and Bell states have an intersection point under the same PT. When $p$ is on the left side of the intersection point, the fidelity of the GHZ state is higher than that of the Bell state. With the increasing number of PT, the intersection point moves to the right side.

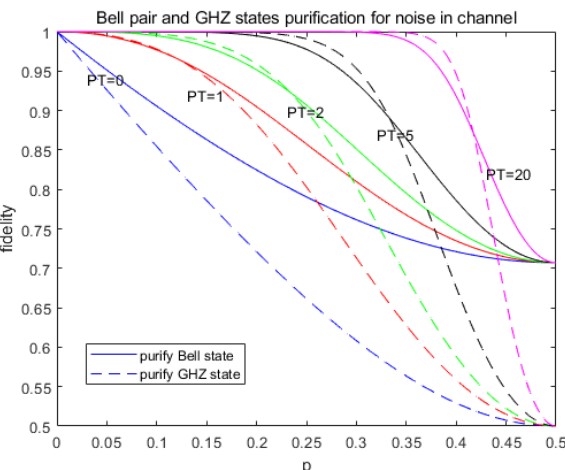

**Figure 9.** Fidelity as a function of the error probability with different PT under bit-flip noise. We use different colors to indicate different purification time, solid line to represent fidelity of Bell states, and dotted line to represent fidelity of GHZ states. The same color represents the same PT. PT = 0 with blue line means that purification is not applied on entangled states. With the increasing number of PT, fidelity of the entangled states will be higher.

We observe that entanglement swapping can bring a 12.77% decline in fidelity. Noise in the quantum channel only affects the entanglement distribution in the network. Purification in the initial entanglement distribution is not so effective because of subsequent entanglement swapping. Thus, when designing the scheme, we focus on the purification after each entanglement swapping. In the end, we add an extra purification for fusion to obtain a higher fidelity. So, we set $m_1 = 1$, and $m_2 = 1$. $m_1$ is the PT for entanglement swapping, $m_2$ is the PT for fusion. The purification algorithm of a multi-hop quantum network is shown in Algorithm 1.

---

**Algorithm 1** Purification Algorithm

---

1: **procedure** PURIFICATION($m_1, m_2, n$) ▷ Swapping round n
2:    **while** $m_2 > 0$ **do**
3:        **while** $n > 0$ **do**
4:            finish entanglement distribution between adjacent nodes
5:            finish i round entanglement swapping
6:            **if** purification condition for swapping is true **then**
7:                finish swapping purification for once
8:                $m_1 \leftarrow m_1 - 1$
9:            **end if**
10:            **if** $m_1 == 0$ **then**
11:                $n \leftarrow n - 1$
12:            **end if**
13:        **end while**
14:        **if** purification condition for fusion is true **then**
15:            finish fusion purification for once
16:            $m_2 \leftarrow m_2 - 1$
17:        **end if**
18:    **end while**
19:    **return** fidelity
20: **end procedure**

---

Since we design the network for teleportation, only the entanglement distribution contains transferring quantum states in the channel. To avoid collision of quantum states, the quantum channel is set to simplex. If node B wants to share the entangled pairs with

node A, it can ask node A for an entangled state through the classical channel. Thus, duplex communication is not necessary in the quantum channel.

### 4.2. Simulation Results

The simulation result of the 5-node network in Figure 5 is shown in Figure 10. We should point out that the simulation is influenced by both channel noise and fidelity loss in entanglement swapping. Entanglement swapping does not transfer qubit in the channel, but it surly brings a huge fidelity loss. Thus, we set the fidelity loss at a fixed value of 12.77% in entanglement swapping. Because of the fixed fidelity loss, the system fidelity cannot reach 1 even without noise in the channel.

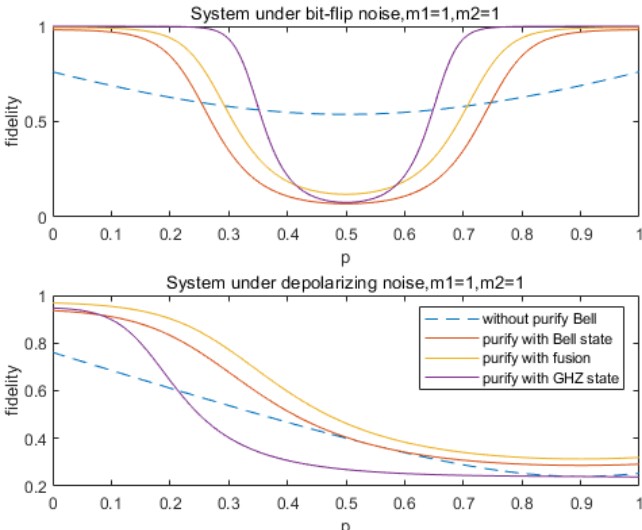

**Figure 10.** Fidelity as a function of bit-flip and depolarizing noise with purification. Purification is applied after every entanglement swapping with time of $m_1 = 1$. After the entanglement is established in the source and distribution node, extra purification is applied with time of $m_2 = 1$.

In Figure 10, the blue line represents the 5-node network based on Bell states without purification. After purification, the orange line represents the Bell states scheme only using Bell states, the purple line represents the GHZ states scheme only using GHZ states, and the yellow line represents the fusion scheme we have proposed. The purification algorithm can improve the fidelity of the final entangled states. When $p$ is small, the fusion scheme and GHZ states scheme are far better than the Bell states scheme under bit-flip noise. However, under depolarizing noise, the advantage of the GHZ states scheme is not obvious, which is even worse than the Bell states scheme most of the time. We can treat depolarizing noise as a mixed noise of bit-flip and phase-flip. In purification of Bell states, there is no effective way to solve the noise with bit-flip and phase-flip together. In addition, in purification of GHZ states, there is no effective way to solve the presence of phase-flip noise. Under bit-flip noise, there is no need to consider the existence of phase-flip noise. Thus, the purification performance under depolarizing noise is most of the time worse than that under bit-flip noise. However, the fusion scheme we have proposed is surely better than the Bell states scheme. The performance of the fusion scheme is close to that of the GHZ states scheme under bit-flip noise. Under depolarizing noise, the fusion scheme is the best.

In the purification scheme we have introduced, each successful purification needs to measure one entangled group. Entanglement swapping also needs to measure lots of entangled states. Purification does not always succeed, which will waste more entanglement. To measure the resource consumption in different schemes, we calculate the minimum basic states required in Algorithm 1. We generate one Bell pair with two basic states based on a H gate and CNOT gate. Although we obtain a GHZ group in the fusion scheme by using two Bell pairs, we use 3 basic states to get the entangled group in the GHZ states scheme.

Considering that measuring only one qubit in the GHZ group leads to the disappearance of the whole entanglement, entanglement swapping in the GHZ states scheme needs to measure two qubits in one group at a time to keep the system entangled. The number of states needed in the network with different node numbers is shown in Table 3. The fusion scheme improves the network performance in comparison to the Bell states scheme and consumes less qubits than the GHZ states scheme.

**Table 3.** Basic states consumed in different schemes, writing as required/measured qubits.

| Node Number | Bell States Scheme | Fusion Scheme | GHZ States Scheme |
|:---:|:---:|:---:|:---:|
| 3 | 16/14 | 20/17 | 36/33 |
| 5 | 64/62 | 68/65 | 216/213 |
| 9 | 256/254 | 260/257 | 1296/1293 |

Simulation is performed for networks spanning different numbers of nodes and is shown in Figure 11. With increasing number of nodes, the fidelity decreases. The performance of the fusion network is significantly better than that of the Bell states scheme.

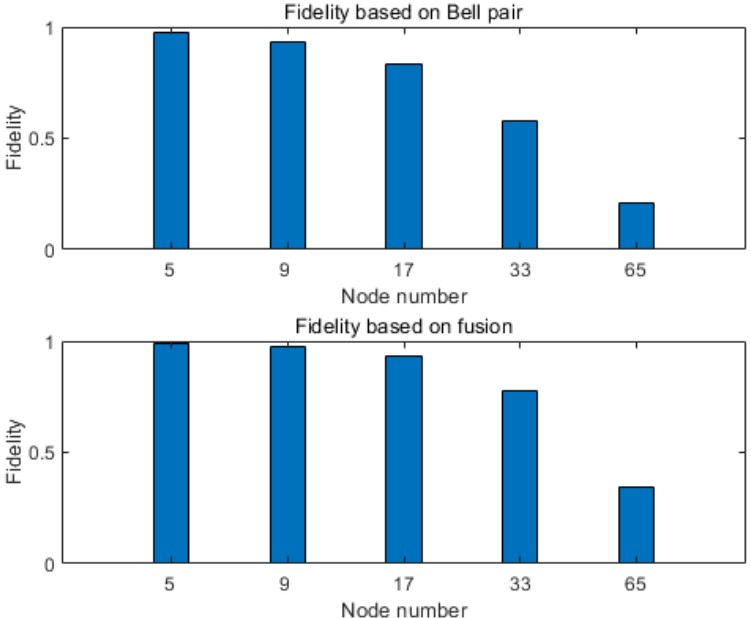

**Figure 11.** The fidelity in simulation of networks with different numbers of nodes in the presence of bit-flip noise, where $q = 0.15$.

In the fusion scheme, the calculation complexity is $O(1)$. After entanglement swapping and purification, high-fidelity entanglement has been established between source and destination node. One can directly use the entangled group to finish teleportation at last. The price is sacrificing a lot of entanglement resources in purification and entanglement swapping.

To verify the scheme in the true environment, we finish the simulation on the NISQ platform Cirq. Here, we choose bit-flip noise and depolarizing noise. Since successful purification is a probabilistic event, not all the simulation results are of reference value. Thus, the consumption is huge. With limited computing power, we set the simulation time for the 3-node network to 5000 and the simulation time for the 5-node network to 1000. The circuit of the 5-node network on Cirq is designed based on the purification algorithm and shown in Figure 12. Simulation results of the 3-node and 5-node network on Cirq are shown in Figure 13.

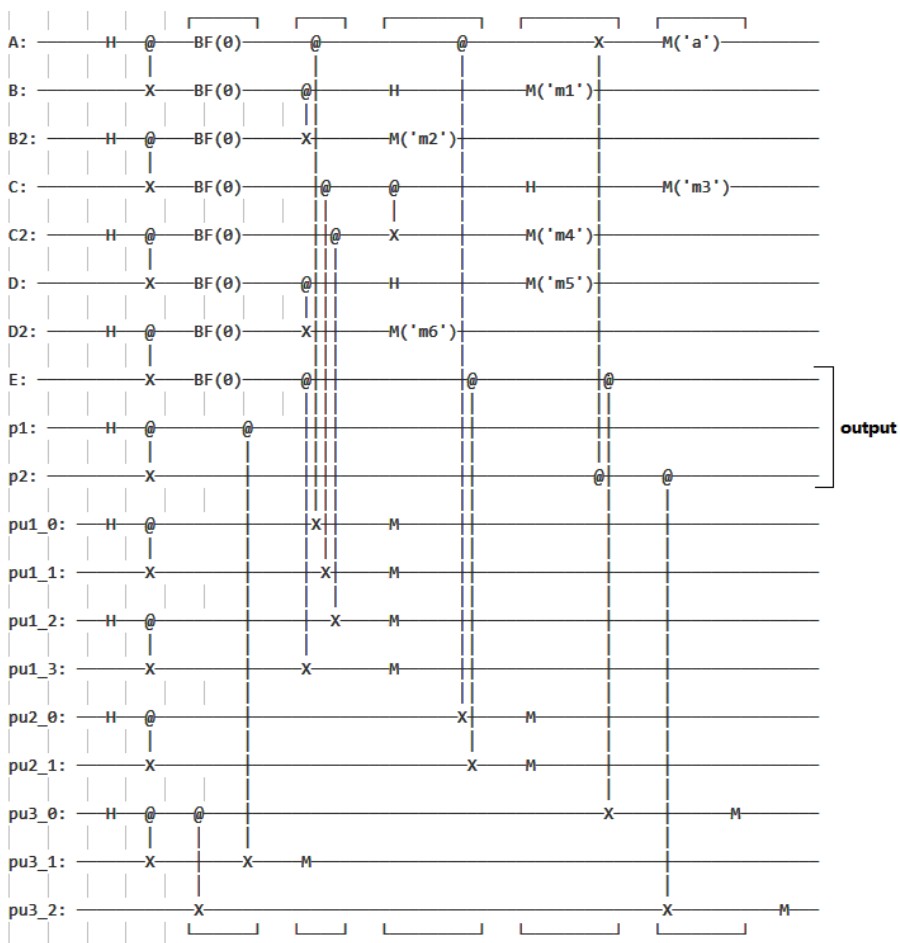

**Figure 12.** The circuit of the 5-node network on Cirq. A, B, B2, C, C2, D, D2, and E are the initial entangled states shared by adjacent nodes. The fusion resources are p1 and p2. Target states in purification are $pui\_j$, where $i = 1, 2, 3$ and $j = 0, 1, 2, 3$. The output of the 5-node circuit comes from states in E, p1, and p2. To avoid the huge circuit, we prepare the target states with fidelity 1 in $pui\_j$. They should have come from entanglement swapping with less fidelity, but the calculation capacity of the lab does not allow us realizing it on the circuit.

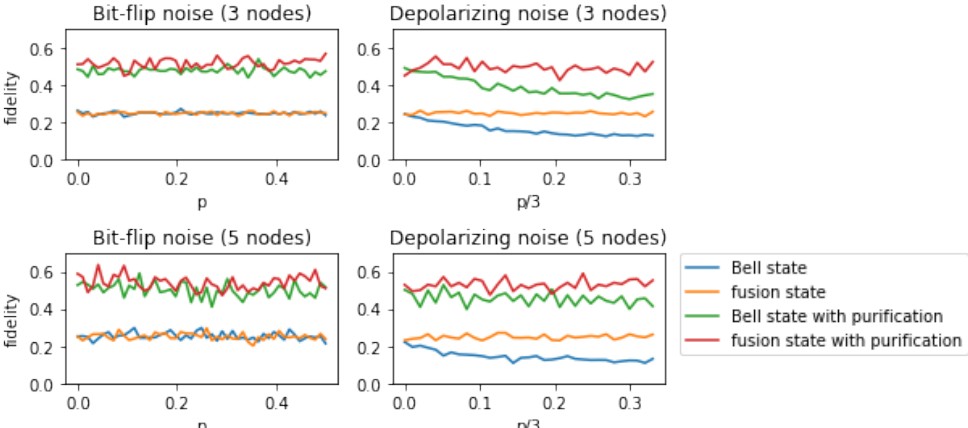

**Figure 13.** The fidelity of the 3-node and 5-node networks under bit-flip noise and depolarizing noise. In each sub-figure, the circuits are the network based on Bell state, the network based on fusion state, and two networks with purification.

The simulation results of the realistic platform can be found to be far worse than the theoretical expectation. When the noise parameter is 0, the fidelity of the network without

purification stays around 0.25 on Cirq no matter how many nodes there are. Entanglement swapping can bring fidelity loss not only because of the noise in the channel. The loss value in entanglement swapping we use comes from the work in [18], which is hard to realize on Cirq. However, we can still observe that the fidelity gain comes from purification. The network based on fusion state shows stronger robustness under depolarizing noise. Since entanglement swapping brings serious loss of fidelity, the influence of bit-flip noise is hard to observe. After purification, the network based on fusion state has higher fidelity than that of the Bell state network under both bit-flip and depolarizing noise.

## 5. Conclusions

In this paper, we propose a novel multi-hop quantum network based on fusion to balance the network performance and consumption. Entanglement swapping can establish entanglement between nodes in multi-hop networks, while fusion can transform entanglement from Bell states to GHZ states. Purification can ensure high fidelity of the entanglement in the network. A purification algorithm is given to ensure the robustness of the communication system. In the algorithm, we set $m_1 = 1$ and $m_2 = 1$, which means applying a one-time purification after each entanglement swapping and a one-time purification after fusion. Simulation results indicate that our new scheme has better robustness against the noise than the Bell states scheme. It also consumes less entanglement than the GHZ states scheme. The calculation complexity is kept at a low level. Moreover, we give the circuit of a 5-node network and simulate it under bit-flip and depolarizing noise on the NISQ platform Cirq to verify our network. Finally, simulation results confirm the superiority of the proposed scheme in balancing network performance and consumption.

Some problems still need to be solved. First, the influence of distance between nodes in the quantum network is not considered. Second, the performance of entanglement swapping can also be improved by developing a new technology. In the end, since purification needs huge amounts of entanglement and often fails, the number of nodes in the simulation is not enough because of the limited computer capacity.

**Author Contributions:** Conceptualization, J.X. and H.X.; methodology, J.X.; software, J.X.; validation, J.X., M.M. and P.W.; formal analysis, J.X.; investigation, J.X.; resources, J.X.; data curation, J.X.; writing—original draft preparation, J.X.; writing—review and editing, J.X.; visualization, J.X.; supervision, J.X.; project administration, X.C.; funding acquisition, X.C. All authors have read and agreed to the published version of the manuscript.

**Funding:** This research received no external funding.

**Institutional Review Board Statement:** Not applicable.

**Informed Consent Statement:** Not applicable.

**Data Availability Statement:** Not applicable.

**Conflicts of Interest:** The authors declare no conflicts of interest.

## Appendix A

Assume that node A wants to send a quantum state $|\psi\rangle$ to node B, the state of the system is $|\Gamma\rangle$. The Bell pair shared between two nodes is $|\Phi^+\rangle$.

$$
\begin{aligned}
|\Gamma\rangle =& |\psi\rangle \otimes |\Phi^+\rangle = (\alpha|0\rangle + \beta|1\rangle) \otimes \frac{1}{\sqrt{2}}(|0_A 0_B\rangle + |1_A 1_B\rangle) \\
=& \frac{1}{\sqrt{2}}[\alpha(|00_A 0_B\rangle + |01_A 1_B\rangle) + \beta(|10_A 0_B\rangle + |11_A 1_B\rangle)].
\end{aligned}
\tag{A1}
$$

$|\psi\rangle$ is the control state, while the state with subscript A is the target state. Node A operates CNOT gate, and then applies the H gate to $|\psi\rangle$. $|\Gamma\rangle$ is transformed to $|\Gamma_1\rangle$.

$$|\Gamma_1\rangle = \frac{1}{2}[|00_A\rangle(\alpha|0_B\rangle + \beta|1_B\rangle) + |01_A\rangle(\alpha|1_B\rangle + \beta|0_B\rangle)$$
$$+|10_A\rangle(\alpha|0_B\rangle - \beta|1_B\rangle) + |11_A\rangle(\alpha|1_B\rangle - \beta|0_B\rangle)]. \tag{A2}$$

Two qubits in node A are measured. Measurement results and the qubit state owned by node B are in Table A1. Quantum gate in node B can rebuild $|\psi\rangle$ with entangled state. Relevant data based on GHZ states are also given in Table A1.

**Table A1.** Teleportation based on Bell pairs and GHZ states.

| States in A | State in B | Operation in B |
| --- | --- | --- |
| $|00_A\rangle$ | $\alpha|0_B\rangle + \beta|1_B\rangle$ | / |
| $|01_A\rangle$ | $\alpha|1_B\rangle + \beta|0_B\rangle$ | X gate |
| $|10_A\rangle$ | $\alpha|0_B\rangle - \beta|1_B\rangle$ | Z gate |
| $|11_A\rangle$ | $\alpha|1_B\rangle - \beta|0_B\rangle$ | ZX gate |
| $|00_A0_A\rangle$ | $\alpha|0_B\rangle + \beta|1_B\rangle$ | / |
| $|01_A1_A\rangle$ | $\alpha|1_B\rangle + \beta|0_B\rangle$ | X gate |
| $|10_A0_A\rangle$ | $\alpha|0_B\rangle - \beta|1_B\rangle$ | Z gate |
| $|11_A1_A\rangle$ | $\alpha|1_B\rangle - \beta|0_B\rangle$ | ZX gate |

**Appendix B**

Both nodes A and C share a Bell pair $|\Phi^+\rangle$ with node B. The system can be described as $|\Lambda\rangle$ .

$$|\Lambda\rangle = \frac{1}{\sqrt{2}}(|0_A0_B\rangle + |1_A1_B\rangle) \otimes \frac{1}{\sqrt{2}}(|0_B0_C\rangle + |1_B1_C\rangle)$$
$$= \frac{1}{2}(|0_A0_c\rangle|00\rangle_B + |0_A1_C\rangle|01\rangle_B + |1_A0_C\rangle|10\rangle_B$$
$$+ |1_A1_C\rangle|11\rangle_B). \tag{A3}$$

Node B operates CNOT gate on the states it has held, and the H gate on the entangled state. The entangled state is the control state, while the other is target state. Now, the system becomes $|\Lambda_1\rangle$ .

$$|\Lambda_1\rangle = \frac{|0_A0_C\rangle + |1_A1_C\rangle}{2\sqrt{2}}|00\rangle_B + \frac{|0_A1_C\rangle + |1_A0_C\rangle}{2\sqrt{2}}|01\rangle_B$$
$$+ \frac{|0_A0_C\rangle - |1_A1_C\rangle}{2\sqrt{2}}|10\rangle_B + \frac{|0_A1_C\rangle - |1_A0_C\rangle}{2\sqrt{2}}|11\rangle_B. \tag{A4}$$

Measurement results in node B and the new entanglement shared by nodes A and C are in Table A2. The quantum gate in node C ensures that the entanglement established is $|\Phi^+\rangle$.

**Table A2.** Entanglement swapping based on Bell states.

| States in B | Shared States in A&C | Operation in C |
| --- | --- | --- |
| $|0_A0_C\rangle_B$ | $|\Phi^+\rangle$ | / |
| $|0_A1_C\rangle_B$ | $|\Psi^+\rangle$ | X gate |
| $|1_A0_C\rangle_B$ | $|\Phi^-\rangle$ | Z gate |
| $|1_A1_C\rangle_B$ | $|\Psi^-\rangle$ | ZX gate |

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
