# Peer review of "A Performance–Consumption Balanced Scheme of Multi-Hop Quantum Networks for Teleportation"

_applsci, doi:10.3390/app112210869_

Round 1
Reviewer 1 Report
No comments
Author Response
Dear reviewer,
Thank you very much for your approval .
Best regards,
Jin Xu
Reviewer 2 Report
Dear Authors,
Thanks for upgrading the paper.
I think you still need to improve section 4. At the moment section 4 contains lots of results and protocol, without explaing and describing them throughly.
Since sections 1-3 is all background, so an expert reader should be able to focus on section 4. My point is section 4 should be self-contained and refrence to previous sections as needed.
Many Thanks
Author Response
Dear reviewer,
Really thanks to your tips. I have revised the section 4 carefully.
Have a nice day.
Best regards,
Jin Xu

Reviewer 3 Report
The authors present a quantum teleportation scheme for multi-hop quantum networks. My highest concern with this manuscript is the presentation of the paper. The manuscript is littered with many ambiguous and technically unsound statements, which hinder my understanding of the technical contribution of the paper. The following are my comments after reading the manuscript thoroughly.
1. Abstract: "Bell state is more susceptible to noise than GHZ state, but using GHZ state consumes more". What do they consume exactly? Please clarify.
2. Introduction: "Teleportation is usually used to send arbitrary state in the quantum system". Teleportation is not 'usually' used, but it is 'always' has been used for transferring quantum information with the aid of maximally entangled states and classical channels. Please revise the statement.
3. Introduction: "But in a multihop quantum network, most nodes are far apart." Please refrain to use an ambiguous word as "far". Two nodes can be physically close to each other with multiple quantum nodes between them, but they also can be physically separated far away but can be directly connected to each other. Please revise the statement accurately regarding what the authors mean by 'far'.
4. Introduction: "To ensure the practicability of the system, we use bit-flip and phase-flip quantum noise model in simulation." If we consider practicability, the quantum photon loss channels may be more practical compared to bit-flip and phase-flip channels. Please clarify.
5. Section II: "Teleportation is a safe communication method." Safe with respect to which aspect exactly? To eavesdropping or to quantum decoherence? Please clarify.
6. Section II: "Teleportation based on GHZ states is shown in Figure 2." Figure 2 does not clearly explain the teleportation mechanism for GHZ states.
7. Section II: "So we choose three most commonly used quantum noise models for channel environment simulation. Consider the complexity of calculation, depolarizing noise is only applied in the simulation." This statement is confusing. So, which channel models are actually being used in the simulation? Please revise.
8. Section III: "Assume that A is the source node while E is the destination node, and they are far apart". Again, please clarify what the authors mean by far?
9. Section III: "Commands can be sent in a classical channel." What commands?
10. Section III: "The principle and circuit of fusion are shown in Figure 4." Figure 4 does not reflect clearly the fusion process of two EPR pairs into a single GHZ state.
11. Section III: It is not clear what is the relation between p and q in Table 1.
12. Section IV: In Figure 9, what are the purification times for GHZ states purification. Are they the same as Bell states? Please clarify. The colour coding can be made identical too.
13. Section IV: "Considering consumption, we set m1 = 1, and m2 = 1. m1 is PT for entanglement swapping, m2 is PT for fusion." I do not understand how the consumption metric works or how it is formally defined in this framework. Why the PT number is also used for fusion while fusion does not perform any purification? Please clarify.
14. Consequently, it is very hard for me now to understand the whole scheme in Figure 10.
15. Section IV: "Thanks to the error correction, assume the influence of noise on the basic state in the quantum frame can be eliminated." This assumption is very strong. It also neglects the fact that GHZ states are much more unstable compared to Bell states.
16. Section IV: "To realize teleportation between node A and E, we need 63 Bell pairs in Bell states 206 scheme, 65 Bell pairs in the fusion scheme, and 126 groups of GHZ states in GHZ states 207 scheme at least." How did the authors come up with these numbers?
17. Section IV: "Calculation complexity is O(2n-1)" and "calculation complexity is O(4)." If the authors are using Big-O notation, normally we neglect the constant factor inside the O notation, so they should be O(n) and O(1), respectively. Also, please clarify, how the authors obtain these complexities.
18. Section IV: The results in Figure 11 will be much interesting if the authors also consider the quantum depolarizing channel model.
In conclusion, the paper may offer an interesting idea, but unfortunately, due to the presentation of the paper, it is hard for me to assess the technical soundness of the paper. Therefore, I'm afraid I cannot accept this paper for publication.
Thank you very much and good luck.
Author Response
Dear reviewer,
Really thanks for your patience and kindly tips. I have revised the manuscript carefully based on your comments.
Have a nice day.
Best regards,
Jin Xu

Round 2
Reviewer 3 Report
Thank you very much to the authors for revising their manuscript according to my comments. The authors include new results, which are portrayed in Fig. 10.
1. In Fig. 10, it is a bit counterintuitive that when the p-value is equal to zero, the fidelity value does not reach 1. The authors must check this result since the only noise is imposed by the channel. Thus, I suspect there are some technical flaws in the derivation of results. Please clarify.
2. The results presented in Fig. 10b suggest that the proposed scheme actually does not introduce an advantage of fidelity improvement when facing quantum depolarizing channels. It is also counterintuitive with the authors' claim that GHZ states are more resilient against noise. Please clarify.
3. The authors should also include the number of auxiliary qubits consumed during the protocol, not just the number of Bell pairs and GHZ pairs since GHZ pairs clearly consume more auxiliary qubits compared to Bell states.
Although, the paper has improved, unfortunately, I still cannot accept this paper for publication in its current form.
Author Response
Dear reviewer,
Really thanks for your patience and tips. I have revised the manuscript carefully. To make the revised part more clearly, I delete the change trace in last version. Changes are mainly in section 4 this time.
Have a nice day.
Best regards,
Jin Xu

Round 3
Reviewer 3 Report
The authors have addressed all my concerns and the paper has significantly improved. I have no further comments.
This manuscript is a resubmission of an earlier submission. The following is a list of the peer review reports and author responses from that submission.
Round 1
Reviewer 1 Report
Dear Authors,
The manuscript written well and quite interesting. I have a few comments
- psi-Quantum company works on fusion-based and entanglement swapping computation. Although your result is not about computation, there are similarities between their work and yours. You might get some advantage by reading their works. Here is an example: https://arxiv.org/abs/2101.09310
- as you have mentioned in the conclusion, your noise model is very simple. I suggest comment more on this matter for realistic noise.
- I believe there are NISQ platforms there you can show the proof of principle of your work
It would be great if you add a section discuss more realistic noise and platform.
Regards
Author Response
Dear Reviewer,
Thanks for your advice.
Best Regards,
Jin Xu

Reviewer 2 Report
1) The captions of the figures are too short. The captions should include
all the information necessary to understand the plots without having to
read an entire section.
2) The conclusions are too concise. It should be convenient to include information about the Multi-Hop protocol that it is improved in the paper
Author Response

(The authors gave the same response as above.)

Reviewer 3 Report
This paper tries to suggest a novel way to implement purification in multi-hop quantum networks with the aim to balance resources required and performance achieved. Unfortunately, it does this in a rather unintelligable way together with an awkward presentation.
The paper starts with the recapitulation of basic concepts (like entanglement, quantum teleportation, noisy channels, fidelity etc) which can be found in any textbook on quantum information, and then focusses on purification issues with emphasis on bit-flip noise in mixed Bell and GHZ states. Up to section 4.1 the paper just presents well-known basic knowledge. In Fig. 10 the cases PT=0 and PT=1 can easily be obtained from the results in 3.2.1. Awkwardly, the formulas given contain various terms which are trivially zero because of orthogonality. The QPA circuit presented in Fig 7 and 8 apperas to be never used. The non-linear map used to obtain PT=2 and higher is not explicitly given.
Despite discussing text-book knowledge this part suffers from poor English and awkward expressions: "entanglement states" instead of "entangled states", "noise chanel" instead of "noisy channel". "Entanglement swapping can bring greater impact compared to the noise in system" What should that mean?? And these are just examples.
The real problem with this paper starts with section 4.2, where the authors try to develop a "Communication protocol based on quantum frame". This section is as mysterious as its title. Shouldn't it mean "Purification protocol... " There is no communication yet as the basic quantum channel is not yet established ! What a cryptic sentence (page 11 second paragraph): "To avoid collision in the quantum channel the entaglement distribution is set to one way transmission" Totally incomprehensible! Does the data frame just contain qbits? or entangled states like |Phi+> ("The last qbit in the frame containes the object of the purification")??? How is Figure 12 produced? On top it is labelled by m2=2 in the caption by m2=1 . Which formulas are used to obtain this plot? How is it possible that at p=0 fidelity is smaller than 1.
How do you obtain that 63 Bell pairs are needed to estableish entanglement (of what)? What does that mean? (a Bell pair is maximally entangled. I just can't make sense of this whole section. There are many more issues which I cannot address in this report.
For all these reasons I recoomend to reject this paper and would ask the authors to write a new paper with a much clearer presentation of their ideas
Author Response
Dear Reviewer,
Thanks for your advice. I hope you could read my paper with more patience. Really thanks.
Best Regards,
Jin Xu

Round 2
Reviewer 3 Report
Unfortunately, the authors misinterpreted my previous comments both in detail as well in general. Let me start with some details:
To my opinion the correct expression is "entangled state" and NOT "entaglement state". The authors made changes into the wrong direction throuout the paper. Furthermore, as was noted in the authors reply that in the cited PRL one speaks of a noisy channel and not a noise channel, which is something different indeed. So in the title of of 2.4 it should read "noisy channel" and then correspondingly in the rest of the paper. Again the authors changed this into the wrong direction due to a misunderstanding of my comments.
In the new version of the manuscript the authors also introduce a "depolarization channel" (section 2.4) which is written in terms of the desity matrix \rho. Unfortunaely the density matrix is not defined in this section but only later in section 2.5. So a careful reader will not immediately understand Eq. (7). Such a disorganized write-up cannot be accepted for a journal article.
Then I criticised sentences like "entanglement swapping can bring greater impact compared to noise in system". The authors explained what they mean in the letter to the referee.
But still, this sentence is neither acceptable English nor really easily comprehensible. What the authors probably mean is "Entanglement swapping reduces fidelity of a state significantly more than noise in the quantum channel". The don't even express in their sentence the impact "on what". Such writing is confusing and not acceptable for a journal article.
When the autors describe their proposed protocols they come up with the following sentences: " The quantum frame contains the command of purification, the number of purification rounds and operation object which is entanglement state . In purification command there are two operation modes. One is to be sacrificed by the measurement, the other is to be stored".
To be honest, while one may guess what the authors mean, this is not an accetable description. This is just hopeless English in confusing words.
I am sorry to say that even so one may guess some sensible ideas behind this swall of words, such a write-up is to my opinion not acceptable for a decent publication. As I said before the authors should write a clear new text eventuelly with the help of an expert text editor fluent in English.
For these reasons I am unable to recommend this paper for publication.
Author Response
Dear review,
Really thanks for your comments. We have revised again based on your tips.
Best Regards,
Jin Xu

Round 3
Reviewer 3 Report
Let me focus on the most important part of the paper, section 4.2
As I pointed out already in my previous reports, the description of the quantum frame is mysterious. See line 288 "For a quantum frame shown in fig. 10 , if we devide a quantum frame into a lot of time slots, we can transmit one state in a time slot". This is just not comprehensible to my opinion.
The authors should precisely define what they mean with "quantum frame" which they actually write as standard quantum state consisting of 4 entangeled qbits |1,2,3,4>. However they are transmitted at different times. Does this mean we have a state with four different time labels |1(t_1),2(t_2),3(t_3),4(t_4)>?? As I said before, the description of the key concepts of the paper (line 288 to 310) is written in a mysterious way in poor English. Their is a lot to say about this text, and until this part is not carefully rewritten, this paper should not be published.
Sentences like "One means it will be sacrified by the measurement, ..." are just hilariously obscure.
I cannot recommend this paper for publication.